# Fabrication of Dragee Containing *Spirulina platensis* Microalgae to Enrich Corn Snack and Evaluate Its Sensorial, Physicochemical and Nutritional Properties

**DOI:** 10.3390/foods11131909

**Published:** 2022-06-27

**Authors:** Maryam Bayat Tork, Mohsen Vazifedoost, Mohammad Ali Hesarinejad, Zohreh Didar, Masoud Shafafi Zenoozian

**Affiliations:** 1Department of Food Science and Technology, Neyshabur Branch, Islamic Azad University, Neyshabur, Iran; bayat.da@gmail.com (M.B.T.); z_didar57@yahoo.com (Z.D.); 2Department of Food Processing, Research Institute of Food Science and Technology, Mashhad P.O. Box 91735-147, Iran; 3Department of Food Science and Technology, Sabzevar Branch, Islamic Azad University, Sabzevar, Iran; mshafafiz@gmail.com

**Keywords:** snack, *Spirulina platensis*, dragee, fortification, D-optimal mixture design

## Abstract

In this work, the possibility of enriching snacks with *Spirulina palatensis* (SP) powder as a dragee was studied. In dragee formulation, the effects of various levels of SP, sunflower oil, NaCl and sour whey powder on sensory, physicochemical and nutritional properties were investigated. The dragee formulation was optimized and the characteristics of the optimal sample were compared with the control sample (containing dragee without SP). The results showed that adding SP increased the flavonoids, total anthocyanin content, vitamins, protein, minerals, essential and non-essential amino acids and fatty acids, including ω3 and ω6, while decreasing the energy intake. Based on the results, the optimal dragee sample was formulated and prepared with a desirability of 0.955. The correlation coefficient indicated that the effective optimization process and the performance of the model were carried out properly. The addition of SP had a significant impact on all color parameters considered by the panelists, and the enriched sample was given a very good taste score (75.10 ± 2.923) and an outstanding overall acceptance rate (91.20 ± 1.549) by the panelists. Although morphological data from scanning electron microscopy showed the distribution of non-uniform SP particles relative to the addition of SP in the extruded product formulation, the preservation of more nutritional properties and the good acceptance of sensory evaluators indicated the success of the application in dragee formulation. Therefore, instead of being utilized in an extruder, we discovered that SP may be used as a dragee for snack fortification.

## 1. Introduction

Given the popularity of snacks and their potential to improve diet quality, enriching them is recommended in order to have a favorable impact on the nutritional pattern of society while also eliminating the harmful effects of snacks. Biologically active compounds are naturally present in microalgae cells that can be used to produce new, healthier foods [1].

The food extrusion process is a High-Temperature Short Time process. This technology leads to the production of a wide range of foods with different shapes, textures, flavors and digestibilities [2,3]. Most extruded foods have high energy value but low nutritional value [4]. As a result, a number of studies have been conducted in order to improve the nutritional and functional features of extruded foods [5,6,7]. Some studies have shown that the extrusion process usually reduces the antioxidant activity and total phenolic content of materials [8,9].

*Spirulina palatensis* (SP) is the biomass of the cyanobacterium Arthrospira platensis; it is considered GRAS [10]. SP, filamentous blue green microalgae contain high nutritional value of essential nutrients such as essential fatty acids, essential amino acids, β-carotene [11], vitamins, fiber [12], phenolic compounds [13], minerals and carotenoids [14]. Its proteins are of high biological value and of good quality [15]. SP has 0.0393 mg/100 g selenium. It is also a powerful antioxidant because it contains high quantities of pigments such as chlorophyll (1.56%) and phycocyanin (14.647%) [1,16,17]. SP boosts the immune system and protects against viral infections [18]. SP algae powder can be an excellent alternative to artificial colors in the food industry while also producing useful and valued food products.

There is more research on the use of SP in food fortification, such as SP-enriched food powder with chocolate flavor [19], dry pasta [20], snacks [21] and snack bars [22].

However, there has been no research into the enrichment of snack dragees with SP in extruded foods as far as we know. The researchers looked into the prospect of supplementing corn flour-based snacks with varied levels of SP powder as a functional food. The impact of varied microalgae powder concentrations, sunflower oil content, NaCl concentrations and sour whey powder (SWP) content on sensory and physicochemical aspects, as well as nutritional values, was investigated and optimized.

## 2. Materials and Methods

### 2.1. Materials

SP was provided as a dry powder by Bushehr Salamat Bakhsh Products Company. Dragees were made with a mixture of SP, SWP powder (prepared by Aseman Dairy Products Company, Neishabour, Iran), sunflower oil (prepared by Goncheh Company, Mazandaran, Iran) and salt (prepared by Golha Company, Mashhad, Iran). Pars Firoozeh Factory (Neishabour, Iran) produced uncoated corn-based snacks with special formulation. The extrusion (MPT-E80 model, Mashhad, Iran) temperature was 125 °C, and the opening matrix diameter was 3 mm. Extruded snacks were stored in polypropylene bags at room temperature for 6 h until dragging.

### 2.2. Coating

The extruded products were poured into a dragee pan (Sepehr Machine, Tehran, Iran), and the intended coatings, containing various amounts of SP, SWP, salt and sunflower oil, were pneumatically sprayed (Table 1). Sixteen snack samples at various dragee formulations were coated at 25% by weight, and after draining were stored in flexible hermetic packages made of polypropylene at 18 °C until characterization of the analyses.

### 2.3. Optimization of Dragee Formulation

Mixture design in Design Expert 10 software was used to determine the optimal sample. Color parameters, sensory evaluation, hardness, total flavonoid compounds (TFC) and total anthocyanin compounds (TAC) were all measured as part of the optimization process.

Methanolic extract of the samples was used for experiments. Five grams of the sample powder was poured into the Falcon and 25 mL of methanol was added to it and stirred with Vertex for 2 min. It was then centrifuged at 15 °C for 5 min at 5000 rpm. Twenty-five milliliters of methanol was added to the remaining precipitate in the Falcon and centrifuged again as before. The obtained extracts were filtered through filter paper and centrifuged again under the same conditions. The resulting solution was then reduced to 50 mL with methanol. Methanolic extract was stored in the freezer at −18 °C until the experiments took place. TFC was measured using a small modification of the Tristantini Budi & Amalia method [23]. To determine the adsorption rate of the snack extract, 0.5 mL of the extract was first mixed with 1.5 mL of distilled water and 0.1 mL of potassium acetate of 1 M, and, after 5 min, 0.1 mL of 10% aluminum chloride and 2.8 mL of distilled water were added, and the adsorption rate of the extracts after 30 min at room temperature at 430 nm was read using spectrophotometer (a JenWay 6305 UV-Vis, Bibby Scientific, Staffordshire, UK) and the quercetin concentration of the extracts was determined using a standard curve (R^2^ = 0.996). Samples were analyzed in three replications.

The TAC was determined using a spectrophotometer (JenWay 6305, UK) and the pH differential method [24]. In this method, two buffer systems were used: potassium chloride buffer 0.025 M with pH = 1 and sodium acetate buffer 4.4 M with pH = 4.5. Then, 400 μL of the extract solution was mixed with 3.6 mL of each of the buffers separately and their adsorptions at two wavelengths 510 and 700 nm were read. Results were expressed as mg of Cyaniding-3-glucoside equivalents (CgE). Analyses were in triplicate.

The sensory analysis was performed with 50 children (up to seven years old) using a 5-point hedonic scale with verbal anchors (1 = dislike a lot, 2 = dislike, 3 = neither like nor dislike, 4 = like, 5 = like a lot). The children tested were completely healthy and had a normal weight. Before the test, a questionnaire was completed by mothers, which included questions on sensitivity to a specific food, drug sensitivity, child age, snack consumption and specific diseases. Also, the parents of the children, by filling out forms, announced that they are aware of the ingredients of this product and agree with their child’s sensory evaluation of this product. Color, taste, smell, crispness and overall acceptance were among the traits that were examined. The index of acceptability (AI) was obtained using Equation (1).
(1)$$\mathrm{AI}\left(\%\right)=\frac{\mathrm{scores}\;\times 100}{5}$$

The snacks’ hardness was analyzed using a texture analyzer TA-XT plus (Stable Micro System Ltd., Surrey, UK) and their hardness was calculated. The maximum force needed for a cylindrical probe with a diameter of 2 mm was measured in Newton, the distance from the sample was 5 mm and the probe speed was 1 mm/s.

Color was analysed using a research stereomicroscope (Olympus, 8ZX16 Co., Tokyo, Japan) imaging system in perfectly uniform light conditions without shading. a* value was used to determine the color changes of the samples in order to optimize where the negative values show the greenness of the samples.

### 2.4. Optimal Sample Experiments

After determining the optimal coating formulation, the best sample was subjected to chemical, physical and microbiological tests. All the experiments in this section were performed with five replications for the optimal sample (O) and the snack with algae-free dragee, the control sample (C).

#### 2.4.1. Radical Scavenging Activity (RSA)

The RSA was measured according to Gabr et al. with some changes [25]. Briefly, 250 μL of each of the O and C were added separately to 3 mL of 60 μM ethanol DPPH solution. After 30 min of storage in the dark, the absorption was read by a spectrophotometer (JenWay 6305, UK) at 517 nm and, at the same time, the absorption of the control sample (3 mL DPPH) was obtained under the same conditions. Percent radical scavenging activity was calculated based on Equation (2).
(2)$$\%AA=\frac{\left(Abs_{control}-Abs_{sample}\right)}{Abs_{control}}\times 100$$

#### 2.4.2. Total Phenolic Content (TPC)

The total phenolic compound content of snacks was determined using Folin-Ciocalteu reagent according to Ismaiel et al.’s method with a few modifications [26]. Briefly, 0.4 mL of 10% Folin–Ciocalteu solution was added to 0.2 mL of the extract. The final mixture was kept in the dark at room temperature for 1 h, after diluting the samples at a ratio of 1:5 for O absorption and C were read at 765 nm. To prepare a calibration curve for measuring phenolic compounds to 0.2 mL of gallic acid (stock) solutions at concentrations of 0–100 mg/mL, 0.4 mL of Folin–Ciocalteu and 0.8 mL of sodium carbonate 10% were added, and the adsorption of solutions was read at 672 nm and the standard curve was plotted. The total phenolic content based on the standard curve (R^2^ = 0.997) of gallic acid (0, 10, 20, 30, 40, 50, 60, 70, 80, 90, and 100 μg/mL) was calculated in terms of GAE/g samples.

#### 2.4.3. Basic Chemical Composition

AOAC method was used to measure the moisture, fat, ash, pH value and protein of the O and C, and the value of carbohydrates was calculated by the difference method [27]. The total energy of the samples was calculated according to Sharoba [17].

#### 2.4.4. Amino Acid Analysis

The amino acid profile of snacks was measured according to Volkmann et al. with a few changes. [28]. Briefly, 0.1 g of the sample powder was digested in 6 N HCl for 24 h in an oven at 110 °C. After dilution in water, 0.5 mL of buffer solution (hydrochloric) and 0.4 mL of OPA (ortho-phthalaldehyde) were added to a 0.5 mL sample solution, and then 10 μL of the sample was injected into the HPLC (AZURA model, KNAUER, Berlin, Germany) machine.

#### 2.4.5. Fatty Acids Composition Analysis

Following the procedure of Otles and Pire, 0.2 g of the fat extracted from the samples was added to 2 mL of 0.5 N potassium methanolic hydroxide and 10 mL of methanol, and the sample was refluxed for 30 min to produce methyl ester, after which the sample was cooled [29]. GC gas chromatography was used to evaluate the fatty acid profile. GC with flame ionization detector; GLC column, SP2560 (column length: 100 m; column diameter: 0.25 mm; film thickness: 0.25 µm), cyanopropylpolysiloxane fused silica capillary (FSC) GLC column (Supelco, Bellefonte, PA, USA); flow rates: hydrogen 0.5 mL/min, make-up (He) 35 mL/min; temperatures: injection 220 °C, detector 220 °C; temperature programming in column, starting temperature, 170 °C; final temperature, 220 °C; rate of temperature increase, 1 °C/min; final time, 10 min; split flow, 50 mL/min; split ratio, 100.0/1; velocity, 15.9 cm/s.

#### 2.4.6. Vitamins Assay

Thiamine (B_1_), riboflavin (B_2_), niacin (B_3_), pyridoxine (B_6_), B12, folic acid, vitamin E, vitamin K and biotin were determined using the HPLC system based on the AOAC method [27].

#### 2.4.7. Minerals Content

The model Atomic Absorption Spectrometer (Analytik Jena AG, Jena, Germany) was used to determine the mineral content of snacks. The wavelengths, gaps and currents used for the six elements were as follows: for the zinc element, 213.9 nm, 0.5 nm and 4 mA; for calcium, 422.7 nm, 1.2 nm and 4 mA; for copper, 324.8 nm, 1.2 nm and 3 mA; for sodium 589 nm, 0.8 nm and 3 mA; for iron 248.3 nm, 0.2 nm and 6 mA; and for potassium 766.5 nm, 0.8 nm and 4 mA. These results were expressed in mg per 100 g of the sample weight.

#### 2.4.8. Scanning Electron Microscopy (SEM)

The microstructures of the O and C samples were studied using a scanning electron microscope (Phenon prox, Eindhoven, The Netherlands) with an accelerating voltage of 15 kV and a working distance of 76.55 mm to 108 μm and magnifications of 40, 500, 2300 and 2500. The samples were sprayed with a palladium–gold combination for 3 min before being examined for morphology.

#### 2.4.9. Color Evaluation

The O images were taken using a research stereomicroscope imaging system (Olympus, 8ZX16) and with three replications. Image J ver.1.4.g was used to evaluate the color of the snacks, and the color specifications of the samples were L* ((black (0)/white (100)), a* ((green (−60)/red (+60)) and b* (blue (−60)/yellow (+60)) was extracted. According to Equation (3), total color changes (ΔE) were calculated, where the values of L, a and b correspond to the O sample and the values of L_0_, a_0_ and b_0_ relate to the C sample. The other color parameters of chroma (C*), hue angle (h) and whiteness index (WI) were calculated using Equations (4)–(6), respectively [30].
(3)$$\Delta E=\sqrt{{\left(\Delta L^{*}\right)}^{2}+{\left(\Delta a^{*}\right)}^{2}+{\left(\Delta b^{*}\right)}^{2}}$$
(4)C*=(a*2+b*2)12
(5)$$h^{0}=\tan^{-1}\left(\frac{b^{*}}{a^{*}}\right)$$
(6)$$\mathrm{WI}=100-\sqrt{{\left(100-L^{*}\right)}^{2}+a^{{*}^{2}}+}b^{{*}^{2}}$$

### 2.5. Optimal Mixture Design

#### 2.5.1. Statistical Model Match

Using the optimization method, the desired values for each variable for the final product were obtained. In this study, 16-Run, 5 factors were investigated by designing D-optimal mixture design to determine the optimal formulation of independent variables and observe the responses. Each of the measured responses (dependent variables) was assigned to special cubic and quadratic models. Analysis of variance was used to determine the significance level of the models. For a more detailed look at the role of mixed parameters in the present study, three-dimensional diagrams of 3D Surface, contour and trace plots were used. According to Equation (7) 1, 2, 3 and 4, respectively, are SP, SWP, oil and salt as independent variables and β1 and βu are the lowest and highest levels of variables, respectively. If β_1_ = β_2_ = β_3_ = β_4_, it is a simple design; otherwise, optimum mixture and in the next stage the responses were determined that included TAC, Color (C), Texture (TX), AI and TFC, and in the next stage the correlation rate and finally the statistical optimization were obtained based on the desirability rate, the desired sample was selected, the total desirability was obtained and the performance was calculated.
(7)$$\sum_{i=1}^{n}\beta_{i}=1{\;\mathrm{and}\;\beta }_{1i}\leq \beta_{i}\geq \beta_{\mathrm{ui}\;}\mathrm{where}\;i=1,2,3,4$$

#### 2.5.2. Preparation of Dragee and Experimental Design

The various dragee formulations are prepared and completely homogenized before usage, and the desired design is constrained by independent factors, as shown in Table 1. The first step towards optimal statistical analysis is selecting a model that best describes and fits the data. Hence, the comparison of the models and the lack of fit (LOF) test were carried out to analyze the results.

#### 2.5.3. Statistical Analyses

Statistical analyses were performed using the statistical SPSS software version 25. A significant level of 0.05 was used in the analysis of variance (ANOVA). Analyses were performed in triplicate, and results were presented as mean ± SD (*n* = 3).

## 3. Results

### 3.1. Experimental Results

A special cubic model was used to describe the effect of each variable on TFC, TAC and TX, while a quadratic model was used to describe the effect of each variable on AI and Color. The significance of *p* values, non-significance of LOF and the proximity of Adj R^2^ and R^2^ to 1 all show that the developed models are significantly more reliable and that they can predict responses satisfactorily. In the following, the effect of each variable on the desired properties will be examined. Table 2 shows the predicted and experimental values for responses.

#### 3.1.1. TFC

The quercetin concentration of the extracts was calculated using the standard curve according to the equation (R^2^ = 0.9964) y = 137.84x + 4.3315. According to Table 3, the independent variables A, B and C, and the interaction of BD, CD, ABC, ABD and ACD had a significant effect on the TFC response (*p* < 0.05). The effect of each parameter on TFC is shown in Figure 1. The addition of SP has an increasing effect on TFC, and when microalgae powder, oil and SWP were applied to the snacks as a coating, the TFC increased (Figure 1B). TFC reduced dramatically when salt was added to the mixture, according to the results. The TFC was also reduced by merely adding SWP. Oil had a positive effect on increasing TFC, which is most likely owing to the presence of flavonoids in sunflower oil. Figure 1B Contour plot indicates various changes in TFC against different parameters. As is seen, the effect of salt is much weaker than other parameters, and the variables of SWP, SP and oil have a stronger effect on increasing TFC. As the amount of oil and SP increased in the same order and the value of SWP and salt decreased, a slight increase was observed, which was most likely due to the presence of sunflower oil and flavonoids such as flavanones, flavones, flavonols and isoflavonoids.

#### 3.1.2. TAC

According to Table 3, the independent variables A, B and C and the interaction of BD, CD, ABC, ABD and ACD had a significant effect on the TAC response (*p* < 0.05). The effect of each parameter on TAC is shown in Figure 2. The amount of total anthocyanin increased with increasing the amount of SP, decreasing the amount of SWP, and decreasing the amount of oil in the snack coating, because this microalgae is rich in anthocyanins [17,20]. Oil along with SP did not have a favorable effect on the amount of TAC. TAC, on the other hand, increased with a decrease in SWP and an increase in SP. It has an additive effect on TAC, and the presence of salt has a negative effect (Figure 2A). The software is accurate and reliable, so it is clear from the validation chart that the model developed is effective and robust in predicting performance.

#### 3.1.3. Sensory Evaluation

According to Table 3, the independent variables A, B and C and the interaction of AB, AC, AD, BD and CD had a significant effect on AI response (*p* < 0.05). Figure 3 illustrates the effect of each parameter on the sensory evaluation scores of the samples. The acceptability of snacks is reduced by increasing the quantity of SP to the maximum, increasing the amount of SWP and decreasing the amount of oil. On the other hand, with a decrease in SP to medium values, decrease in SWP and increase in oil, the acceptability rate increased, and, when SP approached zero, the SWP value decreased and the oil-increased desirability values dropped sharply. According to Figure 3, at very low concentrations of SP, the desirability rate is low, and with an increase in SP to medium concentrations, the desirability rate increases. Finally, at high concentrations of SP, the desirability decreases again. This is the opposite of SWP, meaning that at high concentrations, desirability decreased, at medium concentrations, desirability increased and at low concentrations, desirability decreased again. This would show that the panelists, who included children, did not correlate with the high concentrations of SP, which induced algae and sea flavors and did not like the algae flavor, but in moderate concentrations of algae with moderate percentages of SWP and the effect of its taste on algae taste, the evaluated snacks could attract the attention of panelists and were accepted to the extent that they were more accepted in terms of both taste and color than samples without SP. It might show that children tend to eat snacks of different colors and even new flavors. According to the salt tracing plot, it played an effective role in increasing the desirability of snacks, so that the samples without salt were not very desirable, but the samples accepted by the panelists had lower salt content compared to the snacks available on the market. The panelists probably accepted low salt because of the salty taste of SP, which somehow compensated for the decrease in the amount of salt.

#### 3.1.4. Hardness

Based on Table 3 of the independent variables A, B and C and the interaction of BD, ABC, CD, ABD and ACD had a significant effect on the TX response (*p* < 0.05). The effect of each parameter on the hardness has been shown in Figure 4. Increasing the salt reduced the hardness of the samples. This is probably due to the moisture absorption of salt. Samples containing SP at higher concentrations had lower hardness. In other words, SP has helped to maintain the crispness of the snack texture and has prevented the absorption of moisture from the environment, which ultimately does not cause the texture to soften. This was especially evident in samples where the SWP decreased.

#### 3.1.5. Color

The linear model obtained from ANOVA of the responses was used to evaluate the intensity of the green color. The equation obtained could be used to predict the response for certain levels of each element. By default, the upper levels of the factors are coded as +1 and the lower levels of the factors as −1. The effect of each parameter on the color is shown in Figure 5. Increasing or decreasing SP had a direct and severe effect on the intensity of green color in snacks. The SWP and salt and oil had some effect on changing the color of the snacks. As the SP increased, the intensity of the green color increased. This microalga contains chlorophyll, phycocyanin and β-carotene pigments [11], changing all color parameters, according to research by Lucas et al. (Bárbara Franco Lucas et al., 2018). Hence, in samples that had high levels of SP and less SWP, green color prevailed.

### 3.2. Model Validation

According to the data obtained from the software, the optimal dragged sample was set and prepared with the specified percentages with a desirability of 0.9959 (Figure 6). The predicted and experimental results of model validation based on the optimal formulation are summarized in Table 4. By calculating the difference between the predicted and experimental values, the model error percentage was found to be the highest error percentage between the predicted and laboratory values related to TAC. The purpose of optimization is to find the right set of conditions that meet all goals, not just to reach the desired value of 1 (REF). The empirical models are designed based on the experimental results. The results of the analysis of variance (ANOVA) are shown in Table 3 for the models. It is found that special cubic for TFC, TAC and TX, quadratic model for AI and linear model for color are the most appropriate models.

### 3.3. Optimal Sample Analysis

#### 3.3.1. Chemical and Antioxidant Compounds

The chemical compositions of the O and C were compared in Table 5. Moisture was reduced compared to the C, probably as the combination of SWP with SP reduced the moisture-absorbency of SWP and the final moisture content of the O was reduced compared to the C containing only SWP. The ash content of the O increased by 8% compared to the C and increased significantly (*p* < 0.01). The O had 5.5 times more protein than the C and the protein content increased significantly (*p* < 0.01). The amount of carbohydrates in the C was higher and the value of total energy significantly decreased in the O. The value of TPC was calculated according to the standard curve according to the equation (R^2^ = 0.9978) y = 0.0082 + 0.0334. These values, along with the values of TFC and TAC in the SP, significantly increased compared to the C (*p* < 0.01). The pH of the C was 5.98 ± 0.15 and the O had a pH of 6.85 ± 0.18. This could be due to the high pH of the algae.

Table 6 showed the fatty acid and amino acid profiles of the C and O (*p* < 0.05). The O contained omega 3 and 6 fatty acids as well as all its essential amino acids in significant amounts. Table 7 showed the values of minerals and vitamins in C and O. The O had trace elements such as calcium (936.127 mg/100 g), copper (1.1397 mg/100 g), zinc (3.2314 mg/100 g), sodium (1601.32 mg/100 g, potassium (1994.38 mg/100 g) and iron (269.987 mg/100 g). In the C, compared to the O, only a very small value of calcium, zinc, iron and sodium was detected, which was due to the presence of SWP and salt in the formulation of the snack coating. The C contained vitamins E and B12, whereas the O had high amounts of vitamins. The optimal sample snack contained omega 3 and 6 fatty acids as well as all its essential amino acids in significant amounts. The fatty acid profile obtained and the values of minerals and vitamins detected were in line with the results of some other researchers [16,17].

#### 3.3.2. Physical Tests

##### Microstructure

A scanning Electron Microscope (SEM) was used to collect morphological information. The effects of the addition of the SP were examined using an electron microscope. Figure 7 showed the presence of more air bubbles on the surface of the samples regarding the presence of SP in the coating formulation. As can be seen in Figure 7, in the images of the optimal sample, the addition of SP created grains on the surface of the sample, showing a higher particle density, an effect that was also quite visible on the macroscopic surface. The particle size in the optimum sample was from 895 to 647 nm, showing the hardness of the surface in the images. Given the structure of SP, the distribution of particles was not completely uniform and created relatively irregular particles. In another word, a much more homogeneous microstructure was obtained for the sample without SP with a smoother surface as compared to the irregular and fractured surface shown for the sample containing SP. Unlike Lucas et al. [21,22], who used SP in the extrusion formula, the uniformity of the texture and the uniform distribution of the particles can be seen more in their results.

##### Color Parameters

Figure 8 shows the image of control and optimal snacks and different layers L*, a* and b*. Adding SP significantly affected all color parameters (*p* < 0.05). According to Table 8, L *, b * and C of the O decreased compared to the C, showing the darkness of the color of the O compared to the C, showing that the intensity of the yellow color had decreased, which is because of the presence of chlorophyll pigments, phycocyanins and carotenoids. According to the results of Lucas et al. [21,22], who used SP in the snack extrusion formulation, and Şahin, who used SP and *Dunaliella* in cookie formulation [31], and Olusanya et al. who used *Moringa Oleifera* leaf powder to enrich children’s snacks [32], which is green like SP, the L* component in the O decreased, showing a decrease in the color lightness of the O, which, due to the presence of SP blue–green microalgae, is a change that needs to be fully confirmed.

According to Table 8, the results of the present study were different compared to those of Lucas et al., who used SP in the formulation of snack dough [21], which is probably due to the extrusion of the snack and the pressure and heat applied to them. The color difference between the control sample and the sample containing SP has been greatly increased, and the color difference has been quite obvious and effective. However, according to researchers, values (ΔE) above 6 show a color difference between the samples and the new color of the produced snacks that can be significant for children [33].

The chroma index shows the intensity of color or saturation. According to Table 8, this index decreased due to the darkening of the product color in the O, which was because of the presence of chlorophyll, carotenoid and phycocyanin pigments in the SP. Both Lucas et al. and Sahin faced a decrease in the saturation index [22,31]. The hue angle of 360° or zero shows red, and the closer this angle is to zero, the more intense the red color will be. Angle 90 shows yellow, angle 180 shows green and angle 270 shows blue. The hue angle of the C was reported to be between 90 and 100, which tended to be normal yellow due to the presence of corn flour and SWP. The hue angle of the O was in the green and yellow range, which was probably because of the presence of carotenoids, chlorophyll and phycocyanin pigments in SP. The results of the hue angle parameter were similar to those of Lucas et al. [22]. The whiteness index shows the sample’s tendency to be white, and the closer the value of this component is to 100, the more intense the white color becomes. Based on the data in Table 8, the whiteness index of the C is higher than the O.

##### Hardness

According to the results of De Marco et al. [20], by adding SP to the dragee, the hardness of the snacks increased significantly (*p* < 0.05) and the hardness degree of the sample decreased (Table 9) that is probably because of more porosity and the presence of bubbles. De Marco et al. [20] applied SP to enrich pasta. They reported a higher hardness value in SP-containing pastes compared to the control sample. This contrasts with the results of Lucas et al. [21], who used SP in the extrusion formula and did not report a difference between the hardness of the control sample and SP, which probably could not cause the necessary porosity due to the heat and pressure in the SP extruder.

#### 3.3.3. Sensory Evaluation

Based on Table 9, statistical analysis of sensory parameters (color, flavor, odor, texture and overall acceptance) showed a significant difference between the C and the O (*p* < 0.05). The expected color differences were in the positive direction, and the color of O sample increased the overall acceptance of this sample for most panelists. The texture of the O sample received a higher score (91.80 ± 1.751), confirming the SEM and the textural results. The images of SP powder, the dragee containing it, and the snack coated with this dragee are shown in Appendix A.

## 4. Discussion

Dragee enrichment in extruded snacks will have far higher nutritional properties for those products because the extrusion process is a high pressure and temperature process that can affect the properties of SP. The presence of minerals in the O sample was found to be attributable to the fact that SP is high in minerals such as iron, zinc, copper, sodium, potassium and calcium [17]. SP can greatly boost the quantity of protein in fortified snacks (*p* < 0.01). Due to the lower energy content of O, consuming these snacks compared to market snacks usually does not cause obesity, and the caloric value of consuming this fortified snack will be less for children. SP contains phycocyanin, beta-carotene, xanthophyll, alpha-tocopherol and phenolic compounds, all of which contribute to the alga’s antioxidant properties [34,35].

For this reason, fortified snacks had higher levels of TAC, TFC and TPC. The results show that SP can help keep the body alkaline. Children can safely eat snacks with SP and absorb the nutrients they need [15]. SP is alkaline (pH = 6.93), so the body does not need to excrete calcium to balance the pH, so SP maintains pH. Alkaline blood helps maintain the body’s calcium. Overall, consuming alkaline foods enhances the body metabolic function, strengthens the immune system, proper kidney function and increases the body’s energy level [17]. This microalga is the only complete and harmless source of protein, unlike animal protein sources that increase blood cholesterol [17]. SP is the only food source after breast milk that contains significant amounts of essential fatty acids and γ-linolenic acid that helps regulate the body hormonal system [17].

The results obtained were in line with those of [16,36]. In enriching the pulp of extruded snacks with SP bioactive peptides of LEB 18, da Silva et al. stated that peptides were higher than 4 kDa during the production process of extruded snacks at the peak heat of extrusion (141.33 °C) [5]. They also reported a mass loss for non-hydrolyzed SP. They reported thermal stability for various SP peptides weighing less than and greater than 4 kDa, and they stated that peptides in high-temperature processes SP are not thermally stable and can incorporate. Based on the results of these researchers, the necessity of the present study on the enrichment of coating in extruded snacks will become clearer. The value of vitamin E or tocopherol in this microalga is so high that it extracted this vitamin from SP using the supercritical fluid method [37]. The percentages of SWP and SP formulated for coating could cover the smell of sea and algae well, and, finally, the flavor score with a very good rating (75.10 ± 2.923) was awarded to the O, and its overall acceptance was rated excellent (91.20 ± 1.549) by the panelists. The results showed that the effective optimization process and the performance of the model were carried out correctly. The optimal sample was about 5.5 times more than the control sample containing protein and the amount of protein increased significantly (*p* < 0.01). The ash content of the optimal sample increased more than 8% compared to the control sample and increased significantly (*p* < 0.01), which indicates the presence of minerals in the optimal sample. The amount of total phenolic and flavonoid compounds and antioxidant activity in the enriched snack sample increased significantly (*p* < 0.01) compared to the control sample. There was no significant difference between the sensory parameters in the evaluation of the evaluators, which indicates that the percentages of SWP and SP formulated for draining were able to cover the smell of sea and algae well.

## 5. Conclusions

The present study showed that the *Spirulina platensis* microalgae, when used as a dragee, has a high ability to increase the nutritional value of snacks because it is not exposed to high heat and pressure of an extruder. The results showed that the effective optimization process and model performance were performed correctly. SP supplementation significantly affected all nutritional parameters. Statistical analysis of sensory parameters (color, taste, texture and overall acceptance) showed a significant difference between the control sample and the optimal sample. The color of the final product was more acceptable to the panelists, so the snacks with SP could be effective in promoting people’s health, particularly children and those whose bodies have less capacity to absorb nutrients.

## Figures and Tables

**Figure 1 foods-11-01909-f001:**
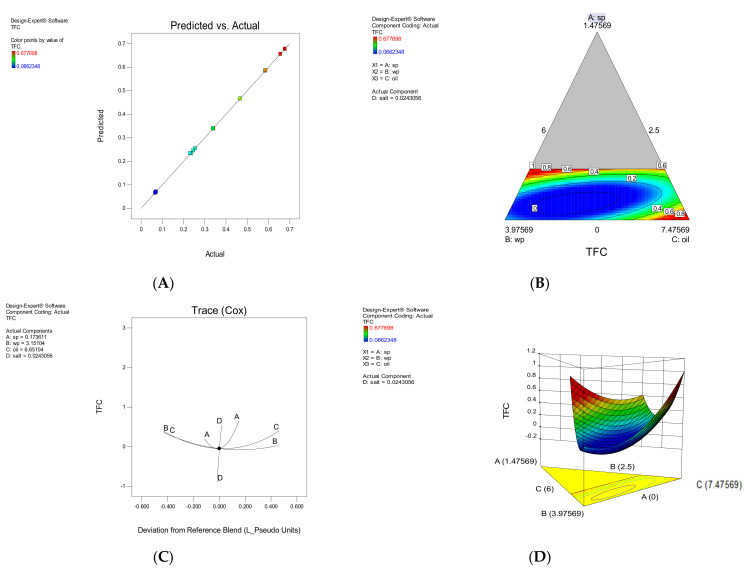
The effect of each parameter on Total flavonoid compounds (TFC), The correlation between experimental and approximated values (**A**), Contour plote (**B**), Trace plot (**C**), response surface plot (**D**).

**Figure 2 foods-11-01909-f002:**
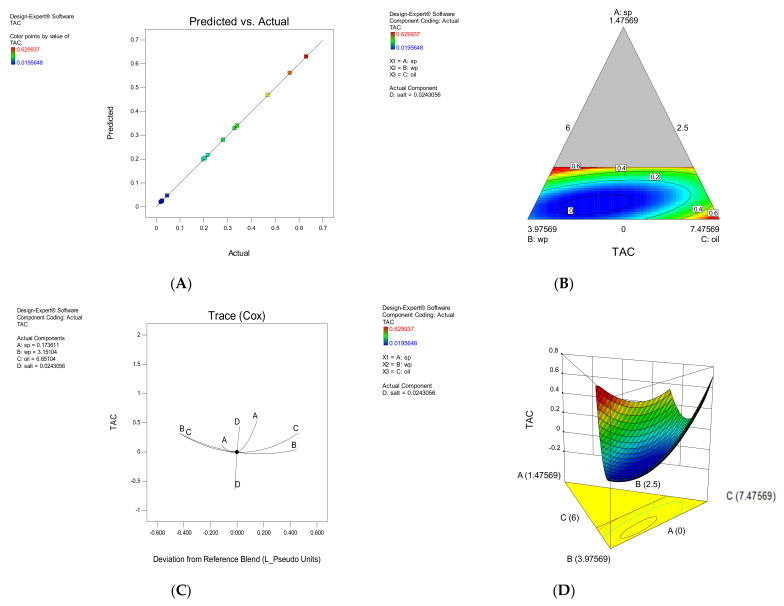
The effect of each parameter on Total anthocyanin compounds (TAC). The correlation between experimental and approximated values (**A**), Contour plot (**B**), Trace plot (**C**), three-dimensional response surface plot (**D**).

**Figure 3 foods-11-01909-f003:**
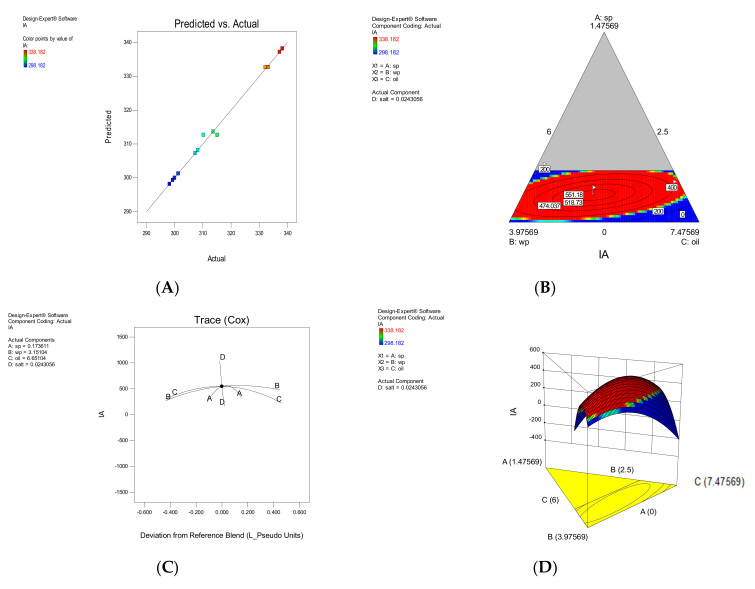
The effect of each parameter on index of acceptability (AI). The correlation between experimental and approximated values (**A**), Contour plot (**B**), Trace plot (**C**), three-dimensional response surface plot (**D**).

**Figure 4 foods-11-01909-f004:**
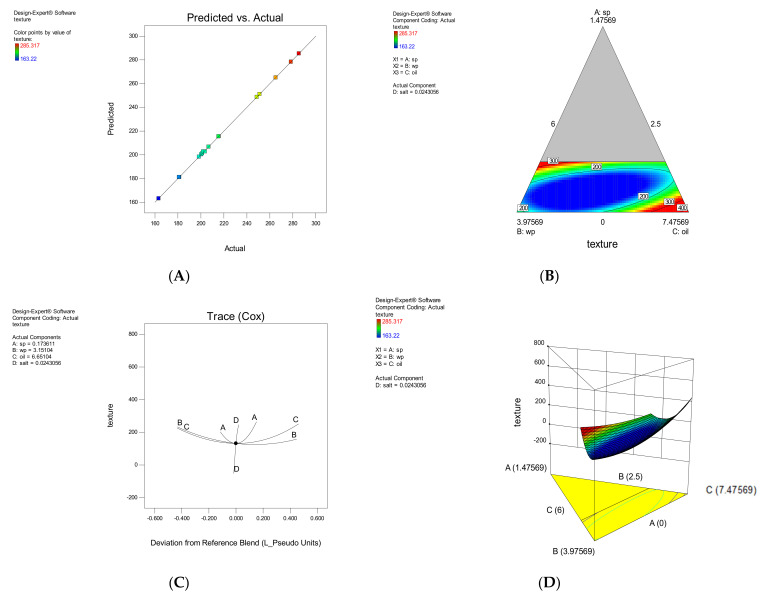
The effect of each parameter on texture (TX). The correlation between experimental and approximated values (**A**), Contour plot (**B**), Trace plot (**C**), three-dimensional response surface plot (**D**).

**Figure 5 foods-11-01909-f005:**
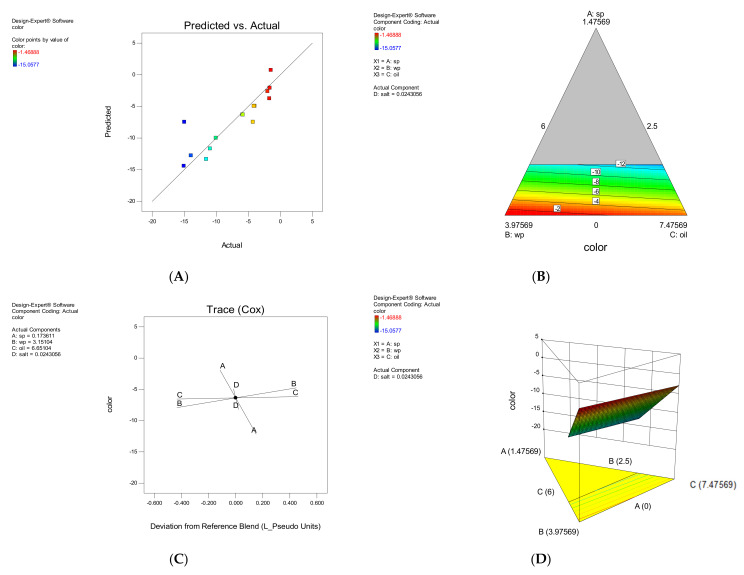
The effect of each parameter on Green color intensity. The correlation between experimental and approximated values (**A**), Contour plot (**B**), Trace plot (**C**), three-dimensional response surface plot (**D**).

**Figure 6 foods-11-01909-f006:**
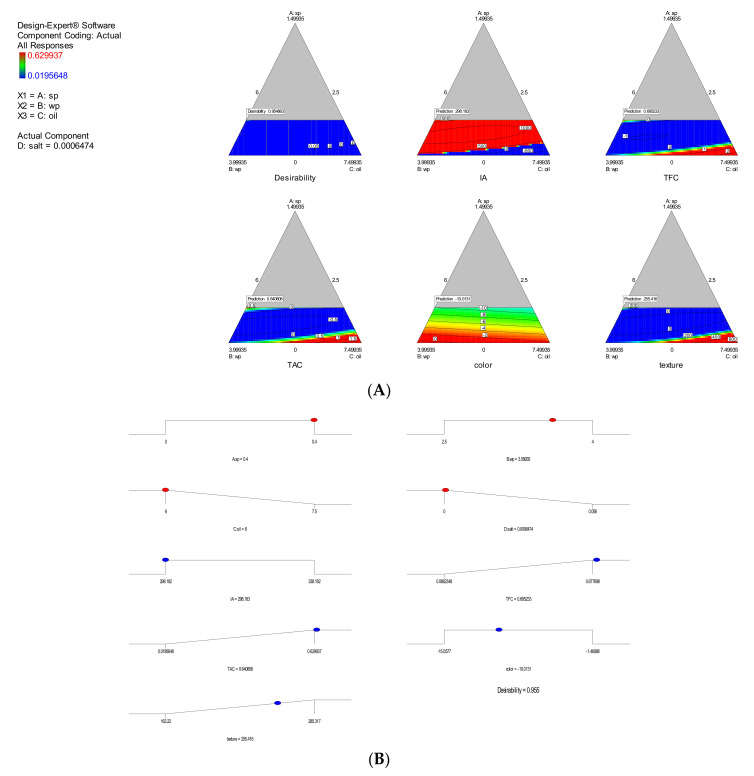
The two-dimensional diagrams of the contour areas (**A**), desirability ramp (**B**).

**Figure 7 foods-11-01909-f007:**
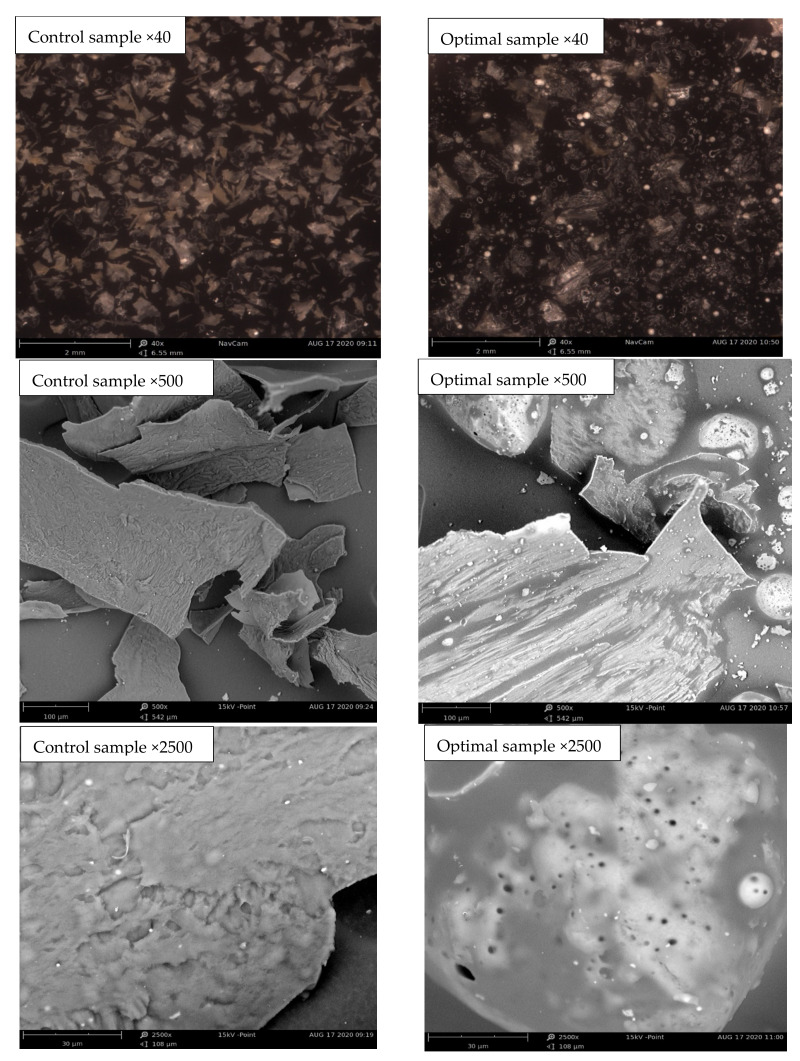
SEM micrographs of the control and optimal samples at different magnifications.

**Figure 8 foods-11-01909-f008:**
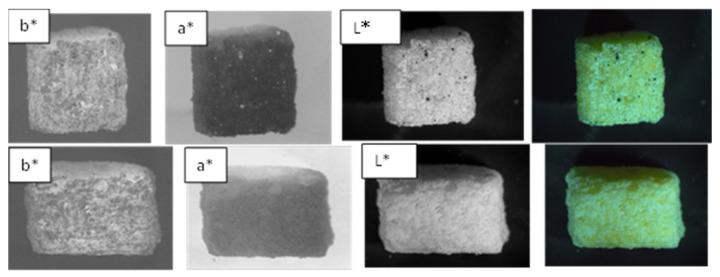
L*, a* and b* of control and optimal snacks (The top row of the optimal sample contains SP and the bottom row of the sample contains no SP).

**Table 1 foods-11-01909-t001:** D-optimal experimental design for the minimum and maximum levels optimization.

Run	SP Powder	SWP	Sunflower Oil	Salt
1	0.00	3.23	6.73	0.02
2	0.13	2.98	6.82	0.05
3	0.13	2.50	7.36	0.00
4	0.40	2.50	7.08	0.01
5	0.13	2.98	6.82	0.05
6	0.40	2.84	6.69	0.05
7	0.10	3.54	6.31	0.04
8	0.20	3.77	6.00	0.02
9	0.00	4.00	6.00	0.00
10	0.40	3.60	6.00	0.00
11	0.00	3.94	6.0	0.05
12	0.40	3.21	6.36	0.01
13	0.40	3.54	6.00	0.05
14	0.00	3.23	6.73	0.02
15	0.00	2.5	7.46	0.03
16	0.20	3.77	6.00	0.02

**Table 2 foods-11-01909-t002:** Predicted and experimental results.

Responses	TFC	TAC	IA	COLOR	TX
predictive value	0.69 ± 0.03	0.64 ± 0.10	298.18 ± 0.25	−10.00 ± 0.04	255.41 ± 0.01
experimental value ^a^	0.70 ± 0.06	0.68 ± 0.12	294.39 ± 0.20	−10.10 ± 0.07	258.97 ± 0.09
%Error ^b^	1.5827	6.5855	−1.2693	1.1185	1.3942

^a^ ±SD Each value is an average of three repetitions. ^b^ %Error = (experimental values–approximated values)/approximated values*100.

**Table 3 foods-11-01909-t003:** Model fitting for each response and the obtained statistical data when applying ANOVA of lack of fit and regression of the selected models.

	Model	F-Value	Model *p*-Value	LOF	LOF *p*-Value	R^2^	Adj R^2^	Predicted Equation
TFC	Special Qubic	9977.74	<0.0001	2.68	0.191	0.999	0.998	+58.32A + 0.070B + 2.68C − 279.41D − 76.33AB − 90.97AC − 2086.90AD − 1.60BC + 290.22BD + 181.96CD + 0.58ABC + 3292.87ABD + 3863.39ACD + 0.000BCD
TAC	Special Qubic	6627.50	<0.0001	2.97	0.171	0.999	0.998	+45.22A + 0.047B + 1.96C − 191.24D − 58.62AB − 68.97AC − 1721.23AD − 1.19BC + 198.07BD + 118.29CD + 0.20ABC + 2633.40ABD + 3046.51ACD + 0.000BCD
IA	Quadratic	55.41	<0.0001	2.85	0.138	0.982	0.978	+362.00A + 311.90B + 276.37C − 52272.41D − 82.73AB + 146.49AC + 54852.71AD − 17.59BC + 54,472.74BD + 54,894.86CD
C	Linear	22.68	<0.0001	1.85	0.987	0.985	0.981	−39.46A + 0.75B − 1.56C − 89.12D
TX	Special Qubic	6118.35	<0.0001	2.63	0.251	0.999	0.999	+11,230.92A + 163.22B + 687.59C − 89,769.18D − 14,642.76AB − 17,612.64AC − 3.249E + 005AD − 514.12BC + 94,476.01BD + 76,190.44CD + 1026.78ABC + 5.800E + 005ABD + 6.746E + 005ACD + 0.000BCD

**Table 4 foods-11-01909-t004:** The optimal sample of D-optimal design.

Number	SP	SWP	Oil	Salt	IA	TFC	TAC	Color	Hardness	Desirability	
1	0.40	3.60	6.00	0.00	308.18	0.67	0.63	−9.97	259.40	0.955	Selected
2	0.36	3.62	6.00	0.01	298.18	0.63	0.53	−9.86	259.71	0.851	
3	0.38	3.31	6.27	0.02	298.18	0.70	0.57	−11.47	246.18	0.805	
4	0.32	3.65	6.00	0.02	298.18	0.58	0.47	−9.33	251.97	0.765	

**Table 5 foods-11-01909-t005:** Chemical composition and physical properties of the optimal (O) and control sample (C).

Parameters	C	O
pH	5.98 ± 0.15 ^b^	6.85 ± 0.18 ^a^
Moisture%	4.38 ± 0.03 ^a^	3.78 ± 0.19 ^b^
Ash%	0.92 ± 0.50 ^b^	8.17 ± 0.22 ^a^
Fat%	1.97 ± 0.10 ^b^	4.74 ± 0.06 ^a^
Protein%	1.42 ± 0.31 ^b^	7.90 ± 0.06 ^a^
Carbohydrate ^a^	91.29 ± 0.29 ^a^	75.39 ± 0.06 ^b^
Energy	487.36 ± 0.47 ^a^	370.84 ± 0.85 ^b^
TPC	0.08 ± 0.03 ^b^	0.55 ± 0.50 ^a^
TFC	0.043 ± 0.04 ^b^	0.69 ± 0.01 ^a^
RSA	0.74 ± 0.54 ^b^	2.85 ± 0.13 ^a^

Different letters in the same line mean significant differences between samples (*p* < 0.05). By difference, values are presented as means ±SD.

**Table 6 foods-11-01909-t006:** Fatty acid and amino acid profiles of the sample of control (C) and optimal snacks (O).

Amino Acids (g/100 g Protein)	Values		Fatty Acids	Values	
Essential amino acids	C	O		C	O
Isoleucine	0.14 ± 0.03 ^a^	6.51 ± 0.05 ^b^	Myristic (C14:0)	12.15 ± 0.02 ^c^	0.42 ± 0.06 ^d^
Leucine	0.53 ± 0.01 ^a^	8.01 ± 0.06 ^b^	Palmitic (C16:0)	14.22 ± 0.01 ^c^	36.98 ± 0.03 ^d^
Lysine	0.12 ± 0.05 ^a^	3.74 ± 0.14 ^b^	Palmitoleic (C16:1 ω 6)	0.006 ± 0.04 ^c^	5.97 ± 0.06 ^d^
Methionine	0.08 ± 0.00 ^a^	2.64 ± 0.01 ^b^	Stearic (C18:0)	3.63 ± 0.14 ^c^	1.96 ± 0.10 ^d^
Phenylalanine	0.21 ± 0.04 ^a^	4.15 ± 0.30 ^b^	Oleic (C18:1 ω 6)	0.03 ± 0.17 ^c^	1.69 ± 0.20 ^d^
Threonine	0.14 ± 0.12 ^a^	3.97 ± 0.13 ^b^	Linoleic (C18:2 ω 6)	1.30 ± 0.51 ^c^	17.01 ± 0.13 ^d^
Tryptophan	0.03 ± 0.30 ^a^	2.01 ± 0.25 ^b^	Gamma-linolenic (C18:3 ω 6)	0.09 ± 0.18 ^c^	26.13 ± 0.50 ^d^
Valine	0.18 ± 0.08 ^a^	5.94 ± 0.17 ^b^	Alpha-linolenic (C18:3 ω3)	0.004 ± 0.04 ^c^	3.44 ± 0.14 ^d^
Total	1.43 ± 0.07 ^a^	36.97 ± 0.04 ^b^	Erucic acid (C22:1)	----	5.12 ± 0.08
Non-essential amino acids			Lignoceric acid (C24:0)	----	1.28 ± 0.36
Alanine	0.44 ± 0.14 ^a^	6.98 ± 0.01 ^b^	Total saturate fatty acid	30.00 ± 0.07 ^c^	40.64 ± 0.05 ^d^
Aspartic	0.37 ± 0.32 ^a^	10.62 ± 0.14 ^b^	Total unsaturated fatty acid	0.13 ± 0.15 ^c^	59.36 ± 0.12 ^d^
Arginine	0.19 ± 0.60 ^a^	7.63 ± 0.20 ^b^			
Cysteine	0.09 ± 0.14 ^a^	1.09 ± 0.17 ^b^			
Glutamic	0.48 ± 0.08 ^a^	13.24 ± 0.13 ^b^			
Histidine	0.12 ± 0.24 ^a^	2.67 ± 0.61 ^b^			
Glycine	0.29 ± 0.21 ^a^	5.12 ± 0.40 ^b^			
Proline	0.45 ± 0.25 ^a^	4.14 ± 0.18 ^b^			
Serine	0.28 ± 0.18 ^a^	4.62 ± 0.27 ^b^			
Tyrosin	0.5 ± 0.05 ^a^	6.92 ± 0.04 ^b^			
Total	3.21 ± 0.21 ^a^	63.03 ± 0.18 ^b^			

Different letters in the same line mean significant differences between samples (*p* < 0.05). Values are presented as means ±SD.

**Table 7 foods-11-01909-t007:** Vitamins and minerals of the sample of control (C) and optimal snacks (O).

Vitamins	mg/100 g		Minerals	mg/100 g	
	C	O		C	O
Vitamin B1(Thiamine)	----	4.60 ± 0.01 mg	Calcium	1.02 ± 0.02 ^b^	936.12 ± 0.16 ^a^
Vitamin B2 (Riboflavin)	----	5.01 ± 0.05 mg	Copper	Trace	1.13 ± 0.13 ^b^
Vitamin B3 (Niacin)	----	15.14 ± 0.03 mg	Zinc	15.00 ± 0.09 ^b^	32.31 ± 0.02 ^a^
Vitamin B6 (Pyridoxine)	----	1.01 ± 0.08 mg	Sodium	1.56 ± 0.46 ^b^	1601.32 ± 41 ^a^
Vitamin B12 (Analogue)	23.80 ± 0.08 ^b^	162.30 ± 0.08 ^a^ μg	Potassium	----	1994.38 ± 0.38
Folic acid	----	9.84 ± 0.08 mg	Iron	13.00 ± 0.32 ^b^	269.98 ± 0.34 ^a^
Vitamin K	----	1102 ± 0.09 μg			
Vitamin E	0.90 ± 0.03 ^b^	8.34 ± 0.12 ^a^ mg			
Biotin	----	7.85 ± 0.61 μg			

Different letters in the same line mean significant differences between samples (*p* < 0.05). Values are presented as means ±SD.

**Table 8 foods-11-01909-t008:** Color parameters of control and optimal snacks.

Parameters	Control	Optimal
L*	32.54 ± 0.11 ^a^	24.65 ± 0.04 ^b^
a*	−13.62 ± 0.01 ^b^	−11.40 ± 0.02 ^a^
b*	9.14 ± 0.08 ^a^	5.60 ± 0.00 ^b^
C	16.40 ± 0.04 ^a^	12.71 ± 0.02 ^b^
h	98.39 ± 0.26 ^b^	121.00 ± 0.50 ^a^
ΔE	-----	8.91 ± 0.073
WI	30.57 ± 0.12 ^a^	23.59 ± 0.04 ^b^

Values are presented as means ±SD (*n* = 10) (*p* < 0.05). Different letters in the same line mean significant differences between samples (*p* < 0.05). L*: Lightness; +a*: redness; −a*: greenness; +b*: yellowness; −b*: blueness; C = Chroma; *h =* hue angle; ΔE: Total color difference; WI = whiteness index; (*): Standard values (L*0, a*0; b*0) used in the calculation of ΔE.

**Table 9 foods-11-01909-t009:** Sensory evaluation and Hardness of control (C) and optimal (O) snacks.

	Flavor	Color	Odor	Texture	Overall Acceptance	Hardness *
C	62.90 ± 1.85 ^b^	57.90 ± 1.96 ^b^	78.80 ± 1.47 ^a^	81.20 ± 1.81 ^b^	82.20 ± 1.98 ^b^	196.71 ± 0.94 ^a^
O	75.10 ± 2.92 ^a^	72.90 ± 2.80 ^a^	78.10 ± 1.72 ^a^	91.80 ± 1.75 ^a^	91.20 ± 1.54 ^a^	134.15 ± 0.98 ^b^

Values are presented as means ±SD (*n* = 10) (*p* < 0.05). * Values are presented as means ±SD (*n* = 3) (*p* < 0.05). Different letters in the same column mean significant differences between samples (*p* < 0.05).

## Data Availability

The data presented in this study are available on request from the corresponding author.

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
