# Peer review of "Fabrication of Dragee Containing Spirulina platensis Microalgae to Enrich Corn Snack and Evaluate Its Sensorial, Physicochemical and Nutritional Properties"

_foods, 2022, doi:10.3390/foods11131909_

Round 1
Reviewer 1 Report
The objective of the study about the enrichment of snacks with various levels of Spirulina palatensis (SP) powder as a dragee could be very interesting. Nevertheless, the manuscript, in this reviewed form, is not well structured and written. Considering this reviewed form, some important modification and implementations are necessary. I hope these followings comments are helpful.
First of all, the study set up in this way is too long and dispersed. I would like to suggest dividing it into two articles where in the first the development of the enriched snack is shown and discussed and in the second the characteristics of the optimal product are studied in-depth and the sensory analysis also showed and discussed.
1. Comments to Author:
The objective of the study about the enrichment of snacks with various levels of Spirulina palatensis (SP) powder as a dragee could be very interesting. Nevertheless, the manuscript, in this reviewed form, is not well structured and written. Considering this reviewed form, some important modification and implementations are necessary. I hope these followings comments are helpful.
First of all, the study set up in this way is too long and dispersed. I would like to suggest dividing it into two articles where in the first the development of the enriched snack is shown and discussed and in the second the characteristics of the optimal product are studied in-depth and the sensory analysis also showed and discussed.
Many sentences are not well constructed in a correct English grammar, for examples lines 52-54 "SP has 0.0393 mg/100g selenium and high quantities of pigments including chlorophyll (1.56%) and phycocyanin (14.647%) is considered a powerful antioxidant". Besides, many paragraphs, dealing with the same topic, are composed of short, fragmented sentences and with several repetitions.
So please re-read the text carefully and try to make the sentences clearer and more fluid.
- MATERIALS AND METHODS:
-Line 83: I suppose the temperature should be “-18 °C” and not “18 °C), so please change it.
-Line 93: write “M” instead of “molar” (as reported in line 101)
-Line141: “In sum, 0.4 ml of Folin-Ciocalteu 10% was added over 3 minutes”. This sentence is not clear, probably not completed (different if compared to the calibration curve preparation)
-Lines 145-148: I could not understand the dilution of the gallic acid, used for the curve calibration because you write “0-500 mg/ml”(line145) and after you report the following dilution” 0, 10, 20, 30, 40, 50, 60, 70, 80, 90, and 100 µg / ml” at line 148. Please, write this part in a better way.
-Line 158: which buffer do you mean? Please, write something more about it
-Line159: What OPA does it mean?
-Line160: please, write some more information about the HPLC method applied.
-Line 165: “The hexane layer was then put into the device after it was extracted with hexane”. This sentence is not clear, what do you mean? Please, write it in a better way.
-Line 166: please, write some more information about the GC method applied.
-Line 190: please, write “(blue (-60) / yellow (+60)” in the correct bracket.
-Table1: please, could you write the values with the same decimals?
- RESULTS
As regards the experimental results (3.1), the graphs in figures 1, 2, 3, 4 and 5 have dimensions and resolution that do not allow the reader to understand the results. Furthermore, there is no detailed description of them in the text. First, it would be necessary to better indicate which are the dependent variables A, B, C and to make more precise references to the various graphs when the results are described and discussed. In this form, paragraphs 3.1.1, 3.1.2, 3.1.3, 3.1.4 and 3.1.5 are not always so clear. So, please try to make this part better. As I suggested before, you could write more extensively and precisely this part making a single paper about the studying and tuning the optimal product.
-Table4: please, could you write the values with the same decimals? I think are not necessary 4 decimals.
-Fig.6: as I write before, also these graphs are not sufficiently described and discussed in the text.
About the “Optimal sample analysis” (3.3) part, please check all the Tables and standardizes the values using the same decimals, especially between the mean value and its standard deviation. Besides, standardize also the “p” of the statistical limit, using capitol or lowercase letter (both in the text and Tables).
-Line 409: “By calculating the difference 407 between total moisture, ash, fat, and protein from, the value of carbohydrates was obtained, which in the C was higher than the value of carbohydrates, which was naturally due to more body moisture, protein, fat, and ash in the sample.” This sentence is not clear, please write it in a better way.
-Tables 6 and 7: why did you have not reported the statistical analysis (ANOVA) for these results? Please, do it also for these values.
-Fig.7: these images are not discussed in depth in the text to be able to understand the differences they report. Please, write this paragraph in a better way.
-Fig.8: these images are not discussed in the text. Please, write this paragraph in a better way.
-“3.3.3 Microbiological examination”: in this form, this paragraph could be also eliminate. It does not give so important information linked with the other analyses.
-Table 9: Please, could you report the statistical analysis (ANOVA) also for these results?
- CONCLUSIONS
The conclusions are in part a repetition of the discussion paragraph with sentences that are not connected to each other and without a well-harmonized speech. Please, re-write them in fewer sentences, with a more generalize conclusion.

Author Response
Dear Editor and Reviewers,
We deeply appreciate your careful reading and thoughtful comments on our manuscript entitled “Fabrication of dragee containing Spirulina platensis microalgae to enrich corn snack and evaluate its sensorial, physicochemical and nutritional properties”.
Thanks for pointing out the mistakes. We have carefully taken your comments into consideration in preparing our revision and hope that the quality of our manuscript would meet the publication standard of Foods. The following summarizes how we respond to reviewers’ comments.
Response to Reviewer 1
Comments to Author:
The objective of the study about the enrichment of snacks with various levels of Spirulina palatensis (SP) powder as a dragee could be very interesting. Nevertheless, the manuscript, in this reviewed form, is not well structured and written. Considering this reviewed form, some important modification and implementations are necessary. I hope these followings comments are helpful.
First of all, the study set up in this way is too long and dispersed. I would like to suggest dividing it into two articles where in the first the development of the enriched snack is shown and discussed and in the second the characteristics of the optimal product are studied in-depth and the sensory analysis also showed and discussed.
Response: We deeply appreciate you for your careful reading and thoughtful comments on our manuscript again. The language of this manuscript was checked by an English-trainer colleague. We have checked throughout the manuscript and made corrections accordingly and hope that the quality of our manuscript would meet the publication standard of Foods. The following summarizes how we respond to reviewers’ comments.
Many sentences are not well constructed in a correct English grammar, for examples lines 52-54 "SP has 0.0393 mg/100g selenium and high quantities of pigments including chlorophyll (1.56%) and phycocyanin (14.647%) is considered a powerful antioxidant". Besides, many paragraphs, dealing with the same topic, are composed of short, fragmented sentences and with several repetitions.
So please re-read the text carefully and try to make the sentences clearer and more fluid.
Response: Thanks for pointing out the mistake. We have made corrections accordingly. SP has 0.0393 mg/100g selenium. It is also a powerful antioxidant because it contains high quantities of pigments such as chlorophyll (1.56%) and phycocyanin (14.647%). Please see the revised manuscript: Page 2, line 51-53.
- MATERIALS AND METHODS:
-Line 83: I suppose the temperature should be “-18 °C” and not “18 °C), so please change it.
Response: Thank you for the constructive comments. A temperature of 18 °C has been selected to store the sample and perform the necessary tests on the snack.
-Line 93: write “M” instead of “molar” (as reported in line 101)
Response: Thank you for the constructive comments. It was corrected.
-Line141: “In sum, 0.4 ml of Folin-Ciocalteu 10% was added over 3 minutes”. This sentence is not clear, probably not completed (different if compared to the calibration curve preparation)
Response: Thank you for the constructive comments. It was corrected.
-Lines 145-148: I could not understand the dilution of the gallic acid, used for the curve calibration because you write “0-500 mg/ml”(line145) and after you report the following dilution” 0, 10, 20, 30, 40, 50, 60, 70, 80, 90, and 100 µg / ml” at line 148. Please, write this part in a better way.
Response: Thanks for pointing out the mistake. It was corrected.
-Line 158: which buffer do you mean? Please, write something more about it
Response: Thank you for the constructive comments. It was corrected.
-Line159: What OPA does it mean?
Response: Thank you for the constructive comments. It was corrected.
-Line160: please, write some more information about the HPLC method applied.
Response: Thank you for the constructive comments. Some useful information was added in this regard.
-Line 165: “The hexane layer was then put into the device after it was extracted with hexane”. This sentence is not clear, what do you mean? Please, write it in a better way.
Response: Thanks for pointing out the mistake. It was corrected.
-Line 166: please, write some more information about the GC method applied.
Response: Thank you for the constructive comments. Some useful information was added in this regard.
-Line 190: please, write “(blue (-60) / yellow (+60)” in the correct bracket.
Response: Thanks for pointing out the mistake. It was corrected.
-Table1: please, could you write the values with the same decimals?
Response: Thanks for pointing out the mistake. It was corrected.
- RESULTS
As regards the experimental results (3.1), the graphs in figures 1, 2, 3, 4 and 5 have dimensions and resolution that do not allow the reader to understand the results. Furthermore, there is no detailed description of them in the text. First, it would be necessary to better indicate which are the dependent variables A, B, C and to make more precise references to the various graphs when the results are described and discussed. In this form, paragraphs 3.1.1, 3.1.2, 3.1.3, 3.1.4 and 3.1.5 are not always so clear. So, please try to make this part better. As I suggested before, you could write more extensively and precisely this part making a single paper about the studying and tuning the optimal product.
Response: Thank you for the constructive comments. Some useful information was added in this regard.
-Table4: please, could you write the values with the same decimals? I think are not necessary 4 decimals.
Response: Thanks for pointing out the mistake. It was corrected.
-Fig.6: as I write before, also these graphs are not sufficiently described and discussed in the text.
About the “Optimal sample analysis” (3.3) part, please check all the Tables and standardizes the values using the same decimals, especially between the mean value and its standard deviation. Besides, standardize also the “p” of the statistical limit, using capitol or lowercase letter (both in the text and Tables).
Response: Thanks for pointing out the mistake. It was corrected.
-Line 409: “By calculating the difference 407 between total moisture, ash, fat, and protein from, the value of carbohydrates was obtained, which in the C was higher than the value of carbohydrates, which was naturally due to more body moisture, protein, fat, and ash in the sample.” This sentence is not clear, please write it in a better way.
Response: Thanks for pointing out the mistake. It was corrected.
-Tables 6 and 7: why did you have not reported the statistical analysis (ANOVA) for these results? Please, do it also for these values.
Response: Thank you for the constructive comments. It was done.
-Fig.7: these images are not discussed in depth in the text to be able to understand the differences they report. Please, write this paragraph in a better way.
Response: Thank you for the constructive comments. Some useful information was added in this regard.
-Fig.8: these images are not discussed in the text. Please, write this paragraph in a better way.
Response: Thank you for the constructive comments. This figure was transferred to the color parameters section.
-“3.3.3 Microbiological examination”: in this form, this paragraph could be also eliminate. It does not give so important information linked with the other analyses.
Response: Thank you for the constructive comments. It was deleted.
-Table 9: Please, could you report the statistical analysis (ANOVA) also for these results?
Response: Thank you for the constructive comments. It was corrected.
- CONCLUSIONS
The conclusions are in part a repetition of the discussion paragraph with sentences that are not connected to each other and without a well-harmonized speech. Please, re-write them in fewer sentences, with a more generalize conclusion.
Response: Thank you for the constructive comments. It was modified.

Reviewer 2 Report
The manuscript is extensive and contains a large number of results. Precisely because of the amount of data, the results and especially the materials and methods are presented rather vaguely and are therefore difficult to follow. Also the materials described provide a lot of insufficient information about the samples and this part needs to be rewritten. All figures and tables are needed to present the results. However, I recommend the authors to organize the results more clearly, reduce the number of paragraphs, and standardize the results depending on the different factors. Based on all this, I recommend major corrections, with following suggestions to improve the quality of the paper.
ABSTRACT
Lines 18-22 - the sentence is too long, please split it into at least two sentences.
Line 22 - it is not clear what type of control sample was used?
Line 25 - since calorie is the old unit for energy intake, instead of calorie, please write energy intake
Line 26 - it should be noted that after sensory evaluation.
Only the results of sensory analysis were noted in the summary, the results of physicochemical and nutritional properties are not mentioned. Please write a brief summary of the results of the physicochemical and nutritional analysis of the snacks.
INTRODUCTION
Too many paragraphs. For example, authors can insert the description of Spirulina palatensis after line 38 (part of the introduction from lines 47-54).
In this form, the introduction does not provide all the necessary data on the topic. For example, the extrusion process can be described in a little more detail. It is suggested to expand the introduction.
MATERIALS AND METHODS
2.1. Materials.
It is not clear from the data on the plant materials used what type of materials were used. For example, is SP (line 71) supplied in dry powder form or? It is also not clear from the snack dragees what type of product it is (line 72). Perhaps the authors can include some photos of these products in the supplemental files to clearly show the materials used.
Line 75- Please indicate the manufacturer of the model that was used for the extrusion process.
Line 76- since the products were stored, you should indicate under what conditions: Temperature, humidity, refrigerated or frozen?
2.2. Coating
Line 81-82 - a table can be used to give a clearer picture of the samples produced, the preparation methods and the different treatments. I suggest that the authors create a table that clearly shows the patterns, the treatments, and the various factors.
2.3 Optimization - the paragraph heading is not self-explanatory and additional description should be added. Also, the other paragraphs 2.3.1.-2.3.4. are unnecessary and can be included in paragraph 2.3.
Line 92 - it is not clear from the description how the snack extracts were obtained for isolation of TFC from them. The process of sample preparation for TFC should be explained.
Line 99- The same comment applies to the analysis of total anthocyanins. Further explanation on sample preparation should be provided.
2.4 Optimal Sample Experiments- it is not clear why other analyzes were performed after the optimal coating was selected- especially TPC analysis, RSA, etc. (paragraphs 2.4.1.-2.4.10.). In addition, all of these paraphrases are unnecessary and can be included in paragraph 2.4 - the title of the paragraph should more clearly explain the analysis it contains.
In addition, it is not clear what type of sample is involved in the "optimal sample (O)" and the "snack with algae-free dragee, the control sample (C)" - further explanation is needed so that the results can be understood in the main text.
Author Response
Dear Editor and Reviewers,
We deeply appreciate your careful reading and thoughtful comments on our manuscript entitled “Fabrication of dragee containing Spirulina platensis microalgae to enrich corn snack and evaluate its sensorial, physicochemical and nutritional properties”.
Thanks for pointing out the mistakes. We have carefully taken your comments into consideration in preparing our revision and hope that the quality of our manuscript would meet the publication standard of Foods. The following summarizes how we respond to reviewers’ comments.
Response to Reviewer 2
Reviewer 2
The manuscript is extensive and contains a large number of results. Precisely because of the amount of data, the results and especially the materials and methods are presented rather vaguely and are therefore difficult to follow. Also the materials described provide a lot of insufficient information about the samples and this part needs to be rewritten. All figures and tables are needed to present the results. However, I recommend the authors to organize the results more clearly, reduce the number of paragraphs, and standardize the results depending on the different factors. Based on all this, I recommend major corrections, with following suggestions to improve the quality of the paper.
Response: We deeply appreciate you for your careful reading and thoughtful comments on our manuscript again. We have checked throughout the manuscript and made corrections accordingly and hope that the quality of our manuscript would meet the publication standard of Foods. The following summarizes how we respond to reviewers’ comments.
ABSTRACT
Lines 18-22 - the sentence is too long, please split it into at least two sentences.
Response: Thank you for the constructive comments. It was corrected.
Line 22 - it is not clear what type of control sample was used?
Response: Thank you for the constructive comments. It was corrected.
Line 25 - since calorie is the old unit for energy intake, instead of calorie, please write energy intake
Response: Thank you for the constructive comments. It was corrected.
Line 26 - it should be noted that after sensory evaluation.
Only the results of sensory analysis were noted in the summary, the results of physicochemical and nutritional properties are not mentioned. Please write a brief summary of the results of the physicochemical and nutritional analysis of the snacks.
Response: Thank you for the constructive comments. Some useful information was added in this regard.
INTRODUCTION
Too many paragraphs. For example, authors can insert the description of Spirulina palatensis after line 38 (part of the introduction from lines 47-54).
In this form, the introduction does not provide all the necessary data on the topic. For example, the extrusion process can be described in a little more detail. It is suggested to expand the introduction.
Response: Thank you for the constructive comments. It was modified.
MATERIALS AND METHODS
2.1. Materials.
It is not clear from the data on the plant materials used what type of materials were used. For example, is SP (line 71) supplied in dry powder form or? It is also not clear from the snack dragees what type of product it is (line 72). Perhaps the authors can include some photos of these products in the supplemental files to clearly show the materials used.
Response: Thank you for the constructive comments. It was modified. Given that we did not have a photo and had to re-prepare the sample for photography, we have no images to present at this short time.
Line 75- Please indicate the manufacturer of the model that was used for the extrusion process.
Response: Thank you for the constructive comments. It was added.
Line 76- since the products were stored, you should indicate under what conditions: Temperature, humidity, refrigerated or frozen?
Response: Thank you for the constructive comments. It was corrected.
2.2. Coating
Line 81-82 - a table can be used to give a clearer picture of the samples produced, the preparation methods and the different treatments. I suggest that the authors create a table that clearly shows the patterns, the treatments, and the various factors.
Response: Thank you for the constructive comments. It has shown in Table 1.
2.3 Optimization - the paragraph heading is not self-explanatory and additional description should be added. Also, the other paragraphs 2.3.1.-2.3.4. are unnecessary and can be included in paragraph 2.3.
Response: Thank you for the constructive comments. It was done.
Line 92 - it is not clear from the description how the snack extracts were obtained for isolation of TFC from them. The process of sample preparation for TFC should be explained.
Response: Thank you for the constructive comments. It was added.
Line 99- The same comment applies to the analysis of total anthocyanins. Further explanation on sample preparation should be provided.
Response: Thank you for the constructive comments. It was corrected.
2.4 Optimal Sample Experiments- it is not clear why other analyzes were performed after the optimal coating was selected- especially TPC analysis, RSA, etc. (paragraphs 2.4.1.-2.4.10.). In addition, all of these paraphrases are unnecessary and can be included in paragraph 2.4 - the title of the paragraph should more clearly explain the analysis it contains.
Response: Thank you for the constructive comments. It was corrected.
In addition, it is not clear what type of sample is involved in the "optimal sample (O)" and the "snack with algae-free dragee, the control sample (C)" - further explanation is needed so that the results can be understood in the main text.
The optimal sample is added in Table 4, and accordingly, the control sample is also a sample that has all the components of the dragee formulation except SP.

Round 2
Reviewer 1 Report
Dear authors,
thank you for your corrections and hope I think that now the quality of the manuscript has improved.
Best regards
Reviewer 2 Report
The authors have significantly improved the quality of the manuscript based on the comments. Some minor corrections are still needed (see below).
2.3. Please expand the title of paragraph Optimization. In this for the title is not self-explanatory. Instead title paragraph Proximate Composition I suggest the title Basic chemical composition.
Author Response
Dear Editor and Reviewers,
We deeply appreciate your careful reading and thoughtful comments on our manuscript entitled “Fabrication of dragee containing Spirulina platensis microalgae to enrich corn snack and evaluate its sensorial, physicochemical and nutritional properties”.
Thanks for pointing out the mistakes. We have carefully taken your comments into consideration in preparing our revision and hope that the quality of our manuscript would meet the publication standard of Foods.